# Intraoperative Ultrasound in Minimally Invasive Laparoscopic and Robotic Pediatric Surgery: Our Experiences and Literature Review

**DOI:** 10.3390/children10071153

**Published:** 2023-06-30

**Authors:** Marco Di Mitri, Eduje Thomas, Annalisa Di Carmine, Ilaria Manghi, Sara Maria Cravano, Cristian Bisanti, Edoardo Collautti, Francesca Ruspi, Chiara Cordola, Marzia Vastano, Simone D’Antonio, Michele Libri, Tommaso Gargano, Mario Lima

**Affiliations:** Pediatric Surgery Department, IRCCS Sant’Orsola-Malpighi Polyclinic, Alma Mater Studiorum—University of Bologna, 40126 Bologna, Italy; edu.thomas92@gmail.com (E.T.); annalisa.dicarmine@gmail.com (A.D.C.); ilaria.manghi5@studio.unibo.it (I.M.); sara-cravano@libero.it (S.M.C.); bisanticristian96@gmail.com (C.B.); edocolla.ec@gmail.com (E.C.); francesca.ruspi12@gmail.com (F.R.); chiaramberle@gmail.com (C.C.); marzia.vastano@icloud.com (M.V.); simone.dantonio@aosp.bo.it (S.D.); michele.libri@aosp.bo.it (M.L.); tommaso.gargano2@unibo.it (T.G.); mario.lima@unibo.it (M.L.)

**Keywords:** ultrasound, US, robotic surgery, laparoscopic surgery, MIS, pediatric surgery, intra-operative ultrasound

## Abstract

Ultrasound (US) is a non-invasive imaging technique frequently used to examine internal organs and superficial tissues, and invaluable in pediatric patients. In a surgical setting, intraoperative ultrasound allows to highlight anatomical structures in detail during traditional open and minimally invasive surgery, thanks to the use of specific probes. In fact, laparoscopic and robotic ultrasonography requires the development of specialized transducers that fit through laparoscopic trocars. In adults, laparoscopic ultrasound is used during cholecystectomy before dissection of the triangle of Calot, to guide liver biopsies and ablation procedures and for the staging of patients with pancreas adenocarcinoma. However, the applications in the pediatric field are still limited. This paper aims to share our preliminary experience with ultra-sound in minimally invasive laparoscopic and robotic pediatric surgery, describing two cases in which intra-operative ultrasound was applied, and to present a review of the literature on the state of the art of the actual uses in pediatric surgery.

## 1. Introduction

Ultrasound (US) represents a mainstay in the assessment of internal organs and superficial tissues. It is a reliable and non-invasive tool that is highly recommended for the study of the pediatric patient, as in this category, the speed of execution and absence of ionizing radiation make it an ideal first-line diagnostic exam. Its major usefulness resides in the assessment of all abdominal organs, as it yields a better resolution compared to adults. In the newborn, it is also possible to analyze the cerebral parenchyma by US, exploiting the windows between the cranial bones.

Recently, technological development has brought advancement in the available US modalities, introducing the possibility of 3D and 4D US. Moreover, specific US probes ideal for an intraoperative setting have been devised. Intraoperative US is a non-invasive technique that permits the highlighting of anatomical structures in detail during both traditional and minimally invasive surgery. In traditional open procedures, intraoperative US may be performed with probes that are also suitable for transcutaneous applications. However, laparoscopic ultrasonography was made possible only thanks to the development of specialized transducers that fit through laparoscopic trocars, thus allowing highlighting of the margins of the structures to be treated. Various applications of laparoscopic US have been described, such as the definition of the biliary anatomy before the dissection of the triangle of Calot during laparoscopic cholecystectomy. It has also been used to detect liver lesions during biopsies or ablation procedures and during staging patients with pancreas adenocarcinoma. However, fewer applications have been reported in the pediatric field. Our institution is equipped with a 4-Way Laparoscopic probe (BK Medica). This paper aims to share our preliminary experience with this laparoscopic ultrasound instrument, describing two cases treated at our Centre, and to present a review of the literature on the state of the art of the current uses of US in minimally invasive pediatric surgery.

## 2. Case Reports

### 2.1. Laparoscopic Lymphadenectomy

#### 2.1.1. Case Presentation

A 7-year-old girl was brought to our attention for suspected tuberculosis with abdominal lymph node involvement. The patient reported a recent trip of 15 days in Morocco. The patient presented to the emergency department with abdominal pain, nausea, and vomit. The blood exam was normal except for an elevation in C-reactive protein (CRP), which was 3.53 mg/dL. Abdominal US showed a retroperitoneal lymph-adenomegaly of 3.6 cm diameter, confirmed subsequently by abdominal magnetic resonance (MR). Considering the clinical conditions and the medical history, quantiferon and Mantoux intradermal reaction tests were performed, both with positive results (Quantiferon test TB1 Ag (CD4+) 9 UI/mL, TB2 Ag (CD4+, CD8+) 9 UI/mL, negative cutoff 1 UI/mL). The patient came to our attention for the surgical excision of the suspected lymph node to perform microscopic investigations. A cultural exam was positive for Mycobacterium Tuberculosis Simplex, and the patient was started on pharmacological therapy with isoniazid, rifampicin, pyrazinamide, *Ethambutol* hydrochloride and Vitamin B6.

#### 2.1.2. Surgical Treatment

Laparoscopic retroperitoneal lymphadenectomy was performed under general anesthesia. The patient was intubated supine, and a rectus muscle nerve block was performed by the anesthesiologist. A 10 mm trocar, adaptable to 5 mm, was placed transumbilically, and a 5 mm 0° lens Storz was used. Two 5 mm trocars were placed under direct camera vision at the right and left flanks. At last, a 5 mm port for a liver retractor was located at the right hypochondrium and was used to suspend the transverse colon. Pneumoperitoneum was induced (10 mmHg, 1 L/min), the retroperitoneal field was exposed, and the retroperitoneal lymph node station was identified (Figure 1). Before opening a window in the posterior parietal peritoneum, we switched the 5 mm camera to the right-side trocar, and we placed the laparoscopic US 10 mm probe in the transumbilical trocar. By US, we identified the lymph node shown by MR and assessed all critical anatomical components, such as the lymph node hilum, the portal vein, the inferior vena cava, and the abdominal aorta (Figure 2). After the margins of the lymph node were fully recognized, a complete dissection with a laparoscopic dissector and LigaSure™ 5 mm 37 cm (Medtronic Italia SpA, Milan, Italy) was performed. Four clips were placed at the node’s hilum before final excision. The specimen was taken out from the umbilical access. All port sites were closed in layers using absorbable sutures. The entire operation was performed without complications and with minimal bleeding. The patient was discharged on V post-operative day (POD).

### 2.2. Robotic Marsupialization of a Splenic Cyst

#### 2.2.1. Case Presentation

An 11-year-old boy was admitted to our department for elective spleen cystectomy. An incidental diagnosis of a splenic cyst of 3 × 3 cm had been made during a thorax MR for myocarditis. We followed the patient by US, highlighting a volumetric increase of the cyst until it was 6 × 6.7 cm over a period of 12 months. Considering possible risks related to an accidental rupture, we decided to perform an elective robotic marsupialization of the cyst. Before surgery, the patient underwent computerized tomography (CT) to define the anatomy of nearby structures, showing a cyst in the middle portion of the spleen in close relationship with the hilum.

#### 2.2.2. Surgical Treatment

Robotic marsupialization of the splenic cyst was performed under general anesthesia. The patient was intubated supine, and a rectus muscle nerve block was performed by the anesthesiologist. A 10 mm trocar was placed transumbilically, and a 10 mm 30° lens was used. Two 10 mm trocars were placed under direct camera vision at the right hypochondrium and hypogastrium. Lastly, a 12 mm Air Seal port was located at the right flank and was used to induce pneumoperitoneum (12 mmHg). The macroscopic limits of the cyst were not clearly visible, so we decided to use a 10 mm laparoscopic US probe to identify the margins of the lesion (Figure 3 and Figure 4). Then, the cyst was punctured where the thickness of the splenic parenchyma was lesser (2 mm), and its liquid content (70 mL) was aspirated. After complete emptying, marsupialization of the splenic cyst was performed using a robotic vessel sealer. During the maneuver, the ultrasonographic probe was inserted multiple times to ensure the exact location of the cyst and vessels not clearly macroscopically visible. Bleeding control was achieved by placing a hemostatic gauze (Tabotamp^®^) where needed. All port sites were closed in layers using absorbable sutures. A tubular abdominal drain was left in place. The entire operation was performed without complications and with minimal blood loss. The patient was discharged in V POD.

## 3. Discussion

The latest technological progress has led to the availability of new technological instruments for pediatric minimally invasive surgery [1]. US is an imaging technique used daily in clinical practice as a first-level exam. In children, US allows investigation of the thorax, the abdomen, the scrotum and soft tissues [2,3]. In the neonatal period, it is also possible to visualize the cerebral parenchyma because of the patency of the cranial sutures. Intraoperative ultrasound (IOUS) was first performed by Schlegel et al. in 1961 to detect kidney stones [4]. In the following years, IOUS was applied to investigate the biliary tree, becoming an accurate alternative to cholangiography. The first to describe IOUS during a cholecystectomy were Knight and Newell in 1963. They performed an open cholecystectomy and applied the probe directly at the level of the gallbladder, bile ducts, porta hepatis, liver, pancreatic area, terminal portion and ampullary region of the bile duct across the second portion of the duodenum [5]. Stiegmann et al. compared the use of laparoscopic intracorporeal ultrasonography (LICU) of the extrahepatic biliary system with cholangiography for the study of ductal anatomy and the determination of the presence or absence of bile duct stones. Berber et al. reviewed the applications of LICU, showing how it enhances the safety of surgery by facilitating dissection, especially in the case of acute inflammation [6]. The introduction of indocyanine green (ICG) provided surgeons with an alternative means for the intraoperative visualization of the biliary tracts. ICG is a water-soluble, tricarbocyanine dye that returns a green light when flashed with infrared light. After intravenous injection, ICG rapidly binds to plasma proteins and, after being metabolized almost exclusively by the hepatic cells, is secreted entirely into the bile. Due to these characteristics, ICG allows surgeons to have a high-resolution vision of biliary structures during cholecystectomy. With respect to the ease of use of echography, cholangiography by ICG needs intravenous injection about 12 h before surgery [7,8,9,10]. In adults, laparoscopic US is used daily to detect liver lesions and to guide biopsy and ablation procedures. Various applications of LICU in urological surgery are described in the literature [11,12,13,14,15,16]. Liu et al. applied laparoscopic US in renal cell carcinoma surgery, showing an important role of LICU in improving the accuracy of tumor identification, resection and mapping of renal blood vessels, especially in the case of endogenous tumors [15]. In agreement with these results, Qin et al. reported their experiences with LICU in a 583-patient series subjected to laparoscopic partial nephrectomy for intrarenal tumors, concluding that LICU may reduce the need for radical nephrectomy in patients with endogenous renal masses. Fazio et al., in their review about the use of LICU in urological surgery, described a total of 50 cases, including 35 partial nephrectomies, cryoablation of 6 renal tumors, 6 radical nephrectomies, 2 perinephric explorations, and 1 resection of a renal artery aneurysm. Their experience confirms the essential role of LICU in defining the anatomy during laparoscopic perinephric exploration, with focus on the use of LICU to detect iceball formation during laparoscopic renal cryotherapy and to evaluate renal perfusion during laparoscopic renal artery aneurysm repair [16]. Contrary to the adult experiences, the application of LICU in pediatric minimally invasive surgery is still limited, and few applications have been described in the literature. McKay et al. described a distal pancreatectomy for a primitive neuroendocrine tumor in a child with MEN-1, in which LICU proved useful for identifying all lesions and showing safe separation planes from the nearby vasculature [17]. We described two applications of LICU in pediatric minimally invasive surgery, confirming the advantages shown in the experiences with adult patients. In particular, LICU was useful for defining anatomical relationships, allowing accurate identification of the walls of an intrasplenic cyst, the pathologic lymph nodes, the lymph-node hilum and the nearby large vessels. These preliminary results suggest we should increase the use of LICU in pediatric minimally invasive surgery.

## 4. Conclusions

The use of ultrasound in pediatric minimally invasive surgery is still limited due to the need for smaller instruments and the relatively high costs. We have provided our preliminary experience with LICU in a pediatric setting, showing advantages such as greater diagnostic and therapeutic accuracy, as well as safety for the patient during the surgical procedure. Multicenter randomized case-control studies will be important for providing stronger scientific evidence for the widespread application of LICU in laparoscopic and robotic pediatric surgery.

## Figures and Tables

**Figure 1 children-10-01153-f001:**
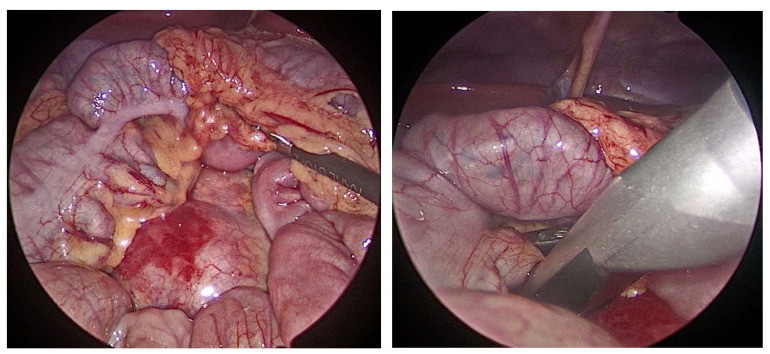
Intraoperative findings of the retroperitoneal lymph node and laparoscopic ultrasound probe.

**Figure 2 children-10-01153-f002:**
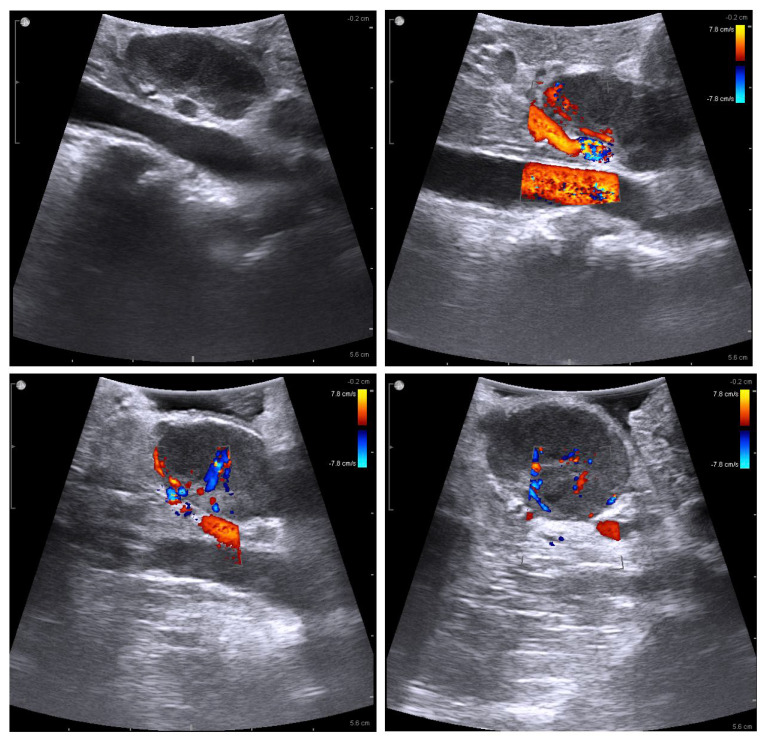
Intra-abdominal laparoscopic US of the retroperitoneal lymph node.

**Figure 3 children-10-01153-f003:**
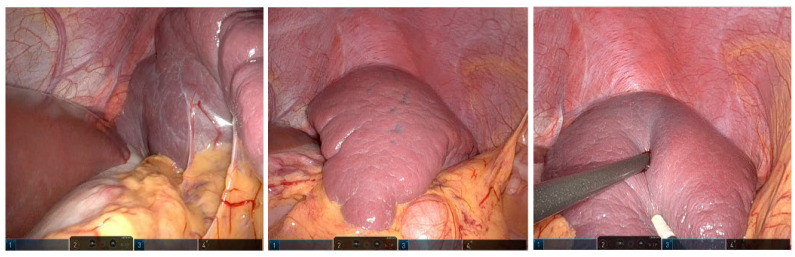
Intraoperative findings of the splenic cyst.

**Figure 4 children-10-01153-f004:**
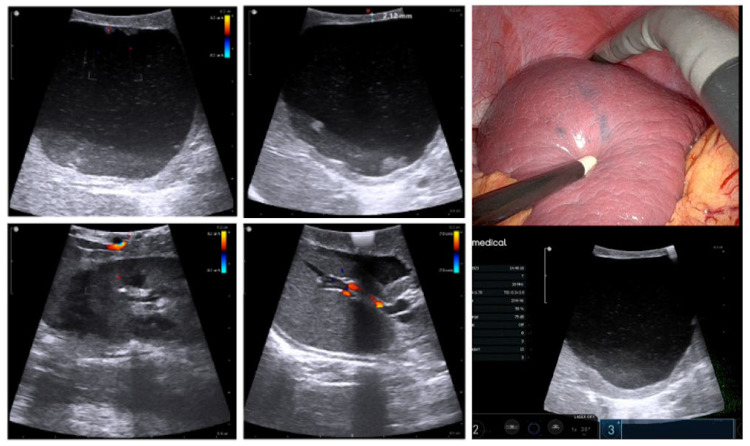
Intraoperative robotic ultrasonography of the spleen cyst before and after the marsupialization.

## Data Availability

The data presented in this study are available on request from the corresponding author. The data are not publicly available due to privacy restrictions.

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
