# Peer review of "Intraoperative Ultrasound in Minimally Invasive Laparoscopic and Robotic Pediatric Surgery: Our Experiences and Literature Review"

_children, 2023, doi:10.3390/children10071153_

Round 1
Reviewer 1 Report
This is a report of two cases whereby the authors used intraoperative ultrasonography (IOUS) in minimally invasive surgery (MIS). The cases are not interesting, and the discussion is just descriptive without any valuable information.
Followings are specific comments.
1. I feel other methods, e.g., transabdominal US or MRI, are sufficient to substitute IOUS in these cases. IOUS should have a significant advantage over other methods to justify making a big incision for using the device.
2. I couldn’t understand how the robot was utilized in the second case.
No problem. I wouldn’t recommend this paper for publication.
Author Response
- The whole purpose of the paper is to stress how useful intraoperative ultrasound can be used to precisely locate lesion or target areas during surgery. Surely, the preoperative diagnostic workup, in most cases, provides sufficient data to know where the target area is located. However, having a direct intraoperative visualization can be priceless in terms of safety and accuracy. Moreover, using the laparoscopic probe requires a 1 cm incision, which certainly does not hamper the final aesthetic result. Our goal is not to prove that intraoperative ultrasound can full replace other diagnostic exams, but only to suggest that it can improve surgery causing no harm to the patient.
- The robotic system was used to perform the marsupialization of the cyst. We have explained better how we used the robot.

Reviewer 2 Report
The authors provide two case reports on the use of laparoscopic ultrsound devices with a review oft he literature to this mentionable topic. Although the review touches an interesting topic it needs some revision.
1. The language needs to be checked for grammar and spelling and style.
Example Page 2, line 71: „referred a recent trip in Marocco“ should be: „reported a recent trip to Marocco“.
The authors should check the manuscript with software (e.g. grammarly) or let it be checked by a native speaker
2. The abstract is identical to the Introduction. Please shorten the abstract and provide only the most necessary aspects of you work.
3. Some of the references are quite old. For example:
„6. Birth M, Ehlers KU, Delinikolas K, Weiser HF. Prospective randomized comparison of laparoscopic 227 ultrasonography using a flexible-tip ultrasound probe and intraoperative dynamic cholangiography during 228 laparoscopic cholecystectomy. Surg Endosc. 1998;12(1):30-36. doi:10.1007/s004649900587“
Although there is some literature on intraoperative US in acute cholecytitis, the technique is not commonly used in simple cholcystlithiasis. Anyhow, if the use improves outcome of patients hast o be questioned and the reference seems outdated.
4. I would like to read about another case. Two cases seem not enough to emphasize the importance of intraoperative ultrasound.
see above.
Author Response
- The manuscript has been reviewed by a native speaker and grammar corrections have been made.
- The abstract has been modified.
- The old references the old references have been deleted
- We only recently acquired the laparoscopic ultrasound, and we are still in the process of collecting ideal cases. However, from this preliminary experience the usefulness and the potential of this instrument seemed quite clear. Surely, we will prepare another paper, in the form of a proper study, once we have enrolled more patients.

Round 2
Reviewer 1 Report
Thank you for the response. Laparoscopic US has been used in lararoscopic live resection and its usefulness has been already shown in adults. But I'm afraid I don't see the advantage of iusing it in the presented pediatric cases at cost of cosmetic results.
No problem
Author Response
Dear Reviewer,
Thank you for reply. We believe that laparoscopic and robotic ultrasound are useful specially to detect vascular structures. Anyway, the cosmetic results are not affected using US because we use a 10 mm umbilical trocar with 5 mm lens and two or three 5mm accessory trocars. When you use the US probe can move the lens into the 5 mm accessory trocars and put the US probe into the 10 mm umbilical trocar.
Reviewer 2 Report
The authors provide a revised version of the manuscript. Unfortunately and although the involved a native speaker, the language still needs improvement (examples are given below). If these rather major issues are addressed the manuscript may be published as a case report but not as an original article because the scientific impact is limited.
Language:
- Discussion: second line of the first paragraph: "Ultrasound is an imaging technique based on ultrasounds ..." .. the wording seems quite odd..
- Later in the discussion, another example, this sentence shows several flaws in spelling, grammar and style: "The showed no statically significant differences between the two techniques, with the only superior of LICU because of the opportunity to repeat the exam infinite times and the potential to reliably major duct injuries because almost all iatrogenic lesions during laparoscopic cholecystectomy are in the suprapancreatic section."
There are other passages with similar problems.. Overall, the language has to be improved. This is not the level of scientific english that is necessary to be published anywhere.
- The abbreviation for ultrasound (US) is made in the introduction. But after that is it hardly used at all...
Language:
- Discussion: second line of the first paragraph: "Ultrasound is an imaging technique based on ultrasounds ..." .. the wording seems quite odd..
- Later in the discussion, another example, this sentence shows several flaws in spelling, grammar and style: "The showed no statically significant differences between the two techniques, with the only superior of LICU because of the opportunity to repeat the exam infinite times and the potential to reliably major duct injuries because almost all iatrogenic lesions during laparoscopic cholecystectomy are in the suprapancreatic section."
There are other passages with similar problems.. Overall, the language has to be improved. This is not the level of scientific english that is necessary to be published anywhere.
Author Response
Dear Reviewer,
Thank you for reply. The manuscript was sent as communication, not as original article.
Probably you read the first version of the manuscript.
After your suggestions, we improved the English language by a native speaker.
For example the phrase "Ultrasound is an imaging technique based on ultrasounds “ it’s not present in the new version of the manuscript.
Anyway, we checked again the English language of the manuscript, and we performed the appropriate corrections.
Round 3
Reviewer 2 Report
The authors addressed the issues. I think the manuscript can be accepted as "communciation" / "case report" or similar.